# The Preparation of Anti-Ultraviolet Composite Films Based on Fish Gelatin and Sodium Alginate Incorporated with Mycosporine-like Amino Acids

**DOI:** 10.3390/polym14152980

**Published:** 2022-07-22

**Authors:** Jing Gan, Chenxia Guan, Xiaoyu Zhang, Lirong Sun, Qinling Zhang, Shihui Pan, Qian Zhang, Hao Chen

**Affiliations:** 1College of Life Science, Yantai University, Yantai 264000, China; ganjing@ytu.edu.cn; 2Marine College, Shandong University, Weihai 264209, China; guanchx@mail.sdu.edu.cn (C.G.); zhangxiaoyucoco@sina.com (X.Z.); 202017661@mail.sdu.edu.cn (L.S.); 202017683@mail.sdu.edu.cn (Q.Z.); panshihui@sdu.edu.cn (S.P.); zhangqianzq@sdu.edu.cn (Q.Z.)

**Keywords:** MAAs, composite film, fish gelatin, sodium alginate, anti-ultraviolet activities

## Abstract

Mycosporine-like amino acids (MAAs) are ultraviolet-absorbing compounds and have antioxidant functions. In this paper, MAAs were added into fish gelatin/sodium alginate films as an anti-ultraviolet additive. The effects of 0–5% MAAs (*w/w*, MAAs/fish gelatin) on the physical properties, antioxidant properties, antibacterial properties and anti-ultraviolet properties of fish gelatin/sodium alginate films were investigated. The results suggest that the content of the MAAs influenced the mechanical properties. The water content, swelling and water vapor permeability of the films were not altered with the addition of MAAs. In addition, the composite films showed effective antioxidant activity and antimicrobial activity. The incorporation of MAAs significantly improved the DPPH radical scavenging activity of the films from 35.77% to 46.61%. Moreover, the block ultraviolet rays’ ability was also greatly improved when the film mixed with the MAAs and when the value of the light transmission was 0.6% at 350 nm. Compared with the pure composite film, the growth of *E. coli* covered by the composite film with 3.75% and 5% MAAs exhibited the best survival rate. These results reveal that MAAs are a good film-forming substrate, and MAAs have good potential to prepare anti-ultraviolet active films and antioxidant active films for applications. Overall, this project provides a theoretical basis for the study of active composite films with anti-ultraviolet activities, and it provides new ideas for the application of MAAs.

## 1. Introduction

In recent years, composite films have received a lot of attention as food or pharmaceutical packaging [1]. In particular, extensive research is now being performed to generate biodegradable packaging from natural biopolymers. Currently, proteins and polysaccharides have been widely adopted to fabricate composite films [2]. Polysaccharide-based films are chemically stable and are based on long-chain helical molecular structures, intramolecular hydrogen robustness and intermolecular hydrogen bonding [3]. Protein-based films have good film-forming properties and remarkable gas barrier properties [4,5]. However, the inferior mechanical properties of the two single network films impose restrictions [6]. Therefore, the blending of polysaccharides and proteins is a valuable strategy to produce composite films with synergistic improved properties. 

Among various biopolymers, fish gelatin is regarded as one of the most excellent packaging materials due to its great film-forming capacity, gas barrier properties and light barrier properties [7,8,9]. However, vulnerability and the lack of sufficient tensile properties seriously hamper the real applications of fish-gelatin-based films [6]. Polysaccharides may be an efficient way to compensate for the shortcomings of fish gelatin films [10]. Pan Ling [11] found that fish gelatin films incorporated with microcrystalline cellulose display more excellent physicochemical properties compared with pure fish gelatin films. Moreover, Yuhao Dong et al. [12] discovered that adding polysaccharides to fish gelatin films dramatically enhances their mechanical characteristics, barrier qualities and tensile strength. Sodium alginates possess high transparency and film-forming capability. However, polysaccharide films prepared from only SA tend to have poor mechanical properties and difficulty meeting the needs of packaging materials [13]. Therefore, the combination of sodium alginate and fish gelatin may be a good choice for the preparation of composite films. The combination of fish gelatin and sodium alginate may be a promising way to overcome their own shortcomings and to reinforce the film properties. 

The incorporation of biologically active substances in composite films can be used to obtain functional diversity of the composite films [14]. Therefore, suitable additives can be the key point for the improvement of composite film performance [15]. Wang and Rhim [7] investigated the influence of composite films incorporated with the extraction of grape seeds on physicochemical properties, and they demonstrated the effectiveness of the composite films on antioxidant activities. Bitencourt [8] and Giménez [16] added turmeric extract to the gelatin film and found that the physical properties, antioxidation and antibacterial properties were improved. For active composite films, antioxidant and antibacterial capabilities have been receiving increased attention. However, a natural active composite film with anti-ultraviolet properties has not been reported yet. The incorporation of bioactive agents into films may be a highly beneficial way to obtain anti-ultraviolet activities.

Prolonged exposure to UV light can cause nutrient loss from food. In particular, certain foods contain vitamins and chlorophyll that are beneficial to humans. However, when exposed to UV radiation, their structure is damaged and degraded. Maria Iliutd [17] found that photochemical breakdown and photosensitization cause riboflavin photodegradation in milk. Moreover, vitamin A is sensitive to UV light. Karla Aguilar [18] demonstrated that ascorbic acid is degraded under UV irradiation. In addition, O Guneser [19] reported that UV light treatment reduces the amount of vitamin B2, vitamin A, vitamin C and vitamin E. Moreover, Huang Chidu [20] suggested that the chlorophyll is broken down during ultraviolet radiation, and UV light accelerates the degradation of chlorophyll. Therefore, there is an urgent need to address the structure of organic molecule destruction problems caused by ultraviolet light. In addition, it is a huge benefit if the packaging material can shield UV rays.

MAAs are natural water-soluble compounds with a low-molecular-weight (<400 Da) [21]. MAAs effectively absorb ultraviolet light, with absorption maxima ranging from 320 to 360 nm [2]. A huge variety of MAAs have been reported in red algae, bacteria, fungi, mollusks, chordates and vertebrates, such as fish [4,5]. The majority of MAA research has concentrated on their photo-protective and antioxidant properties [22]. Moreover, MAAs demonstrate superior antioxidant capability, directly emitting the absorbed radiation by transforming it into heat rather than generating active oxygen [23]. Furthermore, MAAs are excellent candidates in the cosmetics industry for improving skin protection in an eco-friendly manner. With the ability to absorb light in the range of UVA (315–400 nm) and UVB (280–315 nm), MAAs can prevent UVR damage to the skin [24]. M, Sung-Suk investigated the role of MAAs in the human fibroblast cell line HaCaT. He discovered that MAAs, as ultraviolet absorbing compounds, can regulate the expression of genes involved in oxidative stress, inflammation, and skin aging caused by ultraviolet rays [25]. In a nutshell, MAAs are multifunctional natural extracts with anti-ultraviolet, anti-inflammatory and antioxidant properties.

In brief, the focus of this study was to prepare MAAs/fish gelatin/sodium alginate active composite films. To the best of our knowledge, this is the first research on the preparation of anti-ultraviolet packaging films about MAAs. Carrying different concentrations of MAAs, sodium alginate and fish gelatin were used as matrices. Then, the physicochemical properties of the composite edible films, including water content, swelling degree, water solubility, water vapor permeability and rheological properties, were comprehensively evaluated. In addition, whether they had anti-ultraviolet or antioxidation properties was also determined. This project may provide a novel potential candidate material for food preservation and broaden the application of MAAs in the food industry.

## 2. Materials and Methods

### 2.1. Materials

Porphyra was purchased from Rizhao City, Quanzhou City and Ningde City. Fish gelatin was supplied by Vinh Hoan Collagen Corporation (Vietnam), and sodium alginate was a kind gift from Mingyue Seaweed Co., Ltd. All other chemicals were of analytical grade and were purchased from Xinyue Chemical and Glass Co., Ltd. (Weihai, China). *S. aureus* CMCC(B)26003 and *E. coli* ATCC25922 were obtained from Luwei Technology Co., Ltd. (Shanghai, China). All reagents were used without further purification.

### 2.2. Extraction of MAAs

The pretreated porphyra from different regions was ground and placed in 25% ethanol solution (*v*/*v*) at 45 °C for 2 h. After centrifugation (5000 r/min, 20 min, room temperature), the supernatant was evaporated to concentrate under vacuum at 45 °C by rotary evaporation (RE52CS, Yarong Biochemical Instrument Co., Ltd, Shanghai, China) until most of the solvent was removed. Then, the residue was dissolved in ethanol [26], and red or purple red flocculent precipitate was observed. After centrifugation again, translucent pale-yellow supernatant was obtained. Finally, the crude MAAs were dried to a constant weight at 50 °C. The extraction rate (*η*) was calculated using the equation:(1)$$\eta \left(\%\right)=\frac{A}{B}\;*\;100\%$$
where *A* is the weight of MAAs, and *B* is the weight of the porphyra.

### 2.3. Characterization of MAAs

#### 2.3.1. Ultraviolet Spectral Scanning

The dried MAAs were dissolved to 0.1 g/L in distilled water. Following that, the measurement of the maximum absorption wavelengths of the solution in the UV region of 200~400 nm was carried out on a UV visible photometer (T6, Purkinje General Instrument Co., Beijing, China).

#### 2.3.2. DPPH Radical Scavenging Ability

The determination of the DPPH free radical scavenging ability of the MAAs was performed as Kim [27] has described, with slight, minor modifications. Briefly, 1 g/L MAAs was mixed with 0.1 mM DPPH ethanol solution and was placed in a dark place for 30 min. Then, the maximum absorption wavelengths of the DPPH solutions were measured. The measurement was repeated, and each absorbance value was recorded.

### 2.4. Preparation of the Composite Film

The films were formed according to Dou L [28], with some modifications. Firstly, fish gelatin powder was dispersed in distilled water at 60 °C until the fish gelatin was fully dissolved. Next, the sodium alginate powder was added into FG solution with stirring to allow sufficient dissolution. After that, glycerol was added into the FG/SA solution and was stirred at 45 °C for 15 min. In the film solution, the final concentrations of sodium alginate, glycerol and fish gelatin were 1%, 3% and 4% (*w/v*), respectively. To make the anti-ultraviolet films, the MAA solution was added at concentrations of 0%, 1.25%, 2.5%, 3.75% and 5% (*w/w*, named FSM-0, FSM-1.25, FSM-2.5, FSM-3.75, FSM-5.0), and they were mixed for 30 min at 45 °C, followed by degasification for 10 min. Then, the film-forming solutions were cast on a 10 cm × 10 cm glass plate and were dried at 50 °C for 6 h, and then the films were removed from the glass plate and were conditioned at 25 °C and 50% relative humidity (RH) before running further tests. 

### 2.5. Determination of the Physical Properties of the Composite Film

#### 2.5.1. Apparent Surface Color Analysis

The color of the films was measured using a precision colorimeter (NR 110, Shenzhen Sanenchi technology Co., Ltd). The colorimeter was calibrated by a whiteboard (*L**: lightness/brightness, *a**: redness/greenness, and *b**: yellowness/blueness). The chromatic aberrations were calculated using the equation:(2)$$\Delta E=\sqrt{{\left(L-L_{1}\right)}^{2}+{\left(a-a_{1}\right)}^{2}+{\left(b-b_{1}\right)}^{2}}$$
where *L*, *a* and *b* are the color parameter values of film the samples, and *L_1_*, *a_1_* and *b_1_* are the color parameter values of the control film (FSM-0). 

#### 2.5.2. Thickness and Ductility

The average thickness was determined at 5 random positions around the film [29]. The film samples were cut into 4 cm × 1 cm-long strips, and the tensile was tested with a TMS-Pro texture analyzer (TMS-Pro, Petersburg, VA, USA). Elongation at break (EAB) was calculated as follows:(3)$$EAB=\frac{\Delta L}{L0}*100\%$$
where *∆L* is the elongation of the membrane, and *L0* is the initial length of the sample.

#### 2.5.3. Water Content, Swelling Degree and Water Solubility

According to the method of Jamróz E, the test samples, which were 3 cm × 3 cm, were cut from each film, and the initial quality *W_1_* was recorded. Next, the samples were dried (DHG 101-00, Zhejiang, China) at 70 °C for 24 h and were weighed, which was recorded as *W_2_*. Subsequently, the dried films were soaked in 30 mL distilled water at 25 °C for 24 h. The films were taken out, gently blotted with filter paper to remove the surface water and weighed again, and the quality *W_3_* was recorded. Then, the final un-dissolved films were dried at 70 °C for 24 h to a constant weight, and this was determined as *W_4_*. The formulas were calculated as follows [30].
(4)$$Water\;content\left(\%\right)=\frac{\left(W_{1}-W_{2}\right)}{W_{3}}*100\%$$
(5)$$Swelling\;degrees\left(\%\right)=\frac{W_{3}-W_{2}}{W_{2}}*100\%$$
(6)$$Water\;solubility\left(\%\right)=\frac{W_{2}-W_{4}}{W_{2}}*100\%$$

#### 2.5.4. Water Vapor Permeability (WVP)

A glass weighing bottle (5 cm (depth) × 1.5 cm (diameter)) was filled with distilled water (10 mL, 100% RH, 13 °C), and the film was placed over the opening. Then, the bottles were placed in a desiccator containing silica gel (25% RH). The weights of the weighing bottles were recorded at 2 h intervals until the change in weight was constant. The *WVP* (g·mm·m^−2^·h^−1^·kPa^−1^) was calculated by following equation:(7)$$WVP=\frac{\Delta m\times d}{A\times t\times \Delta p}$$
where *Δm* is the quality change in the glass weighing bottle (g), *d* is the average thickness of the membrane (mm), *A* is the film area through which water vapor passes (m^2^), *t* is the time interval (h) and *Δp* is the water vapor pressure difference (KPa) on both sides of the film.

#### 2.5.5. Rheological Experiment

The gelation processes of the film samples were characterized by a rheometer (Haake Mars 3, Thermofisher, Hennigsdorf, Germany) with a rotor (P35TiL) and diameter parallel plate measurement system (diameter: 35 mm, gap 1 mm, shear frequency: 1 Hz, strain: 0.1%). In brief, FSM-0, FSM-1.25 and FSM-5.0 composite solutions (2 mL) were pipetted onto the plate of the instrument. Then, the edge of the geometry was covered with a layer of silicone oil. The temperature increased from 20 °C to 60 °C at a heating rate of 5 °C/min, equilibrated for 15 min, lowered to 10 °C at a rate of 5 °C/min and then equilibrated for another 15 min. The changes in G′, G” and tan δ (G”/G′) were recorded during the whole process [31].

#### 2.5.6. DPPH Radical Scavenging Capacity

Film samples (30 mg) were dissolved in 3 mL distilled water until dissolved completely, and 3 mL of anhydrous ethanol was added. Then, the mixture was centrifuged (10 min, 5000 r) to obtain the supernatant. Supernatant (1 mL) was mixed with DPPH• ethanol solution (5 mL, 0.01 mM) and reacted in the dark for 30 min to measure the absorbance (*A_g_*) at 517 nm. An amount of 1 mL of supernatant and 5 mL anhydrous ethanol solution were mixed, and the absorbance value was *A_c_*. The sample solvent (0.5 mL distilled water and 0.5 mL ethanol) was mixed with 5 mL 0.01 mM DPPH• ethanol solution, and after 30 min of reaction in the dark, the absorbance value was measured as *A_0_*. The radical scavenging activity was calculated as follows: (8)$$Scavengingeffect\left(\%\right)=\frac{A_{o}-A_{g}-A_{c}}{A_{o}}*100\%$$

#### 2.5.7. Antibacterial Properties

The antibacterial efficacy of the composite film against *E. coli* (ATCC25922) and *S. aureus* (CMCC(B)26003) was evaluated according to the inhibition zone. The composite films were cut into 6 mm-diameter discs and were sterilized by ultraviolet light for 20 min. Afterwards, 0.1 mL inoculums containing approximately 10^8^ CFU/mL of bacterial was seeded on the solid LB medium. Then, the composite films were attached to the LB medium after the inoculation and were cultured at 37 °C for 24 h, and the blank filter paper was used as a control. Each film sample was repeated 3 times, and the diameters of the inhibitory zones (mm) were measured.

#### 2.5.8. Optical Blocking Performance

The film sample was cut into a strip with a size of 40 mm × 10 mm and was stuck to the inner wall of one side of the quartz cuvette. An ultraviolet-visible spectrophotometer was used to measure the transmittance of the sample film in the wavelength range of 200~800 nm [32].

Escherichia coli was inoculated in LB liquid medium at 37 °C for 24 h. The diluted bacterial solution (0.1 mL) was taken and spread evenly on the plate, and then each composite film sample was covered on the outside of the plate. Then, FSM-0, FSM-1.25, FSM-2.5, FSM-3.75 and FSM-5.0 were covered on the outside of the plates. The uncoated film plants were used as the control groups. Next, the plates were irradiated under UV light for 15 min and were transferred to a constant-temperature incubator for 24 h at 37 °C. At the same time, plates without UV irradiation were used as positive controls. After 24 h, the growth of the colony was observed, and the appropriate dilution plate was selected for counting.

### 2.6. Statistical Analysis

All experiments were carried out in triplicate to find the mean and standard deviation values. All diagrams were generated with Origin Pro 8.6 software. In addition, statistically significant differences between the groups were used for multiple comparisons by Duncan analysis in SPSS statistic 19.0. Differences were statistically significant at *p* < 0.05.

## 3. Results and Discussion

### 3.1. Physicochemical Properties of MAAs

As shown in Figure 1a, the maximum absorption wavelengths of the three kinds of porphyra extractions were all at 336 nm, which are consistent with the characteristic absorption peaks of MAAs [33]. In addition, the extraction rates of porphyra, Rizhao porphyra, Quanzhou porphyra and Ningde porphyra were, respectively, 6.60%, 7.93% and 6.38%. The highest extraction rate of Quanzhou porphyra was 7.93%. The scavenging capacity of DPPH is one of the most commonly used methods to determine the antioxidant capacity [34]. Interestingly, in Figure 1b, it can be seen that the clearance rate of Ningde porphyra (79.28%) was significantly higher than that of Quanzhou porphyra (50.42%). The results show that the antioxidant effects of the MAAs extracted from the three types were somewhat different. It may be related to the process technology of porphyra [35]. In summary, Quanzhou porphyra had the highest extraction rate and the lowest antioxidant properties. On the other hand, the higher extraction rate and highest antioxidant properties of Ningde porphyrins were beneficial for the subsequent experiments. Thus, Ningde porphyrins were used as raw materials for the subsequent experiments.

### 3.2. The Physicochemical Properties of Composite Films

#### 3.2.1. Surface Color Analysis 

The appearance of food packaging and consumer acceptance can be affected by the color of the edible film to a certain extent, so the color of the edible film is often used as an important evaluation index of the film properties [36]. The transparency of the film facilitates the consumer to observe the morphology of the food contents and the changes that occur during storage. As shown in Figure 2, all films were transparent. Interestingly, the color of the films became pale yellow upon the addition of the MAAs, and the FSM-5 films showed the deepest color and the lowest brightness.

As showed in Table 1, after the addition of MAAs, the L value of the films decreased, and the values of a and b increased, corresponding to the Figure 2 conclusion. The MAAs are light reddish brown after being reconstituted with distilled water. Therefore, the variance in the color parameters may appear to be the consequence of the natural slight yellow of the MAAs.

#### 3.2.2. Thickness and Mechanical Properties

The elongation at break (EAB) can be used to depict the deformation ability of the film. Figure 3 shows that the composite film exhibited higher deformation ability with the addition of MAAs. It may be due to the entanglements and physical-crossing of MAAs, polysaccharides and proteins by inter- and intra-molecular hydrogen bonding interactions, which increased the capabilities of the carrying force [37]. However, the thickness of each composite film sample was about 0.11 mm (*p* > 0.05), and the values had no significant differences among the sample. It may be that the MAAs are water-soluble small-molecule amino acids.

#### 3.2.3. Water Content, Solubility, Swelling Properties and WVP

The water content of the film is the macroscopic characterization of the total water molecules in the film system [38]. The solubility, swelling and WVP of the film were the important criteria for evaluating the water sensitivity of packaging materials, especially in humid environments. The water content (more than 30%) is detrimental to the storage of most food products. As shown in Table 2, the water content values were all below 10%, which were conducive to food storage. Solubility increased from 24.36 ± 5.25% to 32.90 ± 1.42% with the growth of MAA concentrations. It can be seen that the film can be degraded; thus. the films had the advantages of application [39,40]. Moreover, there was no obvious difference in the water content, swelling and WVP values among all composite membranes when the MAAs were added. In general, the results suggest that the composite films with low water content and WVP and with high solubility and swelling properties can be removed from food or drugs easily, which makes them a promising applicable packaging biomaterial for food or drugs.

#### 3.2.4. Rheological Temperature Scanning 

As shown in Figure 4a, in the heating process, the G′ and G″ values of the samples were both continuously decreasing, but the decline of the G″ value was larger. Under the same temperature, the G″ value was always higher than the G′ value. It was indicated that the sample was in a viscous-solution state during the heating process. As shown in Figure 4b, G′ and G″ showed an increasing trend in the cooling process. When the temperature dropped to about 10 °C, the G′ value of each sample increased sharply and exceeded the corresponding G″. It was shown that the gel system gradually transformed into elastic gel. Therefore, it was easy to conclude that the addition of MAAs made the viscosity and elasticity of the composite films slightly reduce.

Combined with the change curve of tan δ (G″/G′) in Figure 4c, it can be seen that FSM 0 was the first to gel (in the cooling stage, tan δ = 1), and the gelling temperature was about 12 °C. It was slightly higher than the gelling temperature of the other samples, which was about 10 °C. However, in the last equilibrium stage at 10 °C, the tan δ curves of each sample overlapped. It may be that the MAAs had good fluidity and could easily be filled in the network structure of the gel. MAAs could provide a number of hydroxyl groups, which could easily form hydrogen bonds with FG molecules and SA molecules. Generally, more intermolecular interactions contributed to the increase in the density of physically cross-linked points, which slightly extended the film-forming time. Moreover, the addition of the MAAs can improve the mechanical properties of the prepared composite films, which is consistent with the measurement results of the tensile properties.

#### 3.2.5. Antioxidant Activity

The DPPH radical scavenging assay was performed to evaluate the antioxidant effects of the films [41]. As shown in Figure 5, the addition of the MAAs significantly increased the DPPH radical scavenging capacity of the composite films. It may be related to changes in the amino acid composition in the system [9]. Moreover, we found that FSM-0 also had antioxidant activity, which was mainly due to the antioxidant activity of fish gelatin and sodium alginate [40]. It was speculated that some amino acid residues in the polypeptide chain and the hydroxyl group in the polysaccharide have the ability of hydrogen donating. Moreover, MAAs also had antioxidant activity, which may be related to changes in the amino acid composition in the system [9]. In general, a more remarkable antioxidant power can be obtained with increasing amounts of MAAs and can further prov the main function of MAAs in the antioxidant activity of the composite films.

#### 3.2.6. Antibacterial Properties 

The composite films have antibacterial power to minimize bacterial growth on food or drug surfaces. Figure 6 shows the antibacterial activity of the composite films against *E. coli* and *S. aureus*. With the increase in MAA content, the area of the inhibition zones increased. In addition, the area of the inhibition zones against *E. coli* and *S. aureus* increased from 8.83 mm and 8.53 mm to 10.94 mm and 10.83 mm, respectively. Overall, the ability of the composite films against *E. coli* was better than the inhibition of *S. aureus,* which may be related to the differences in the cell structure and chemical composition between these two bacteria [42]. As small molecule substances, MAAs were more likely to diffuse on the surface of the bacterial films to achieve an antibacterial effect.

#### 3.2.7. Optical Blocking Performance 

The transmittance of the composite films in the visible area (400~800 nm) was essentially the same as FSM-0, as indicated in Table 3. However, the transmittance decreased with increasing MAA content, indicating that the UV blocking ability of the composite films was good in the ultraviolet region (200~400 nm). It may be that the conjugated double bonds and active groups on distinct side chains in the MAAs boosted the UV absorption capacity of the composite films at specific wavelengths. Moreover, the excellent UV blocking properties of the composite films were also related to the absorption of light by the peptide bond carbonyl group (190~210 nm) in fish gelatin [43]. 

In addition, we measured the light transmittance of FSM-0 and FSM-5 under natural conditions for 7 days (Figure 7a, b). The results show that the light barrier performance of the composite films did not change significantly, indicating that the composite film samples can exist stably under natural conditions.

Furthermore, the ultraviolet resistance of the composite film was demonstrated by bacterial studies. As shown in Figure 8, the plates without the film were exposed to ultraviolet radiation irradiated for 15 min. The *E. coli* colonies were smaller and extremely sparse. On the other hand, the *E. coli* colonies were dense and well-developed on the plates without UV irradiation. Interestingly, when the MAA concentrations were low (less than 2.5%), the *E. coli* colonies were small and uneven. However, as demonstrated in plates 4 and 5 in Figure 8, when the concentrations of the MAAs were increased to 3.75% and above, the number of colonies grew dramatically, and the *E. coli* growth was comparable to that of the non-UV irradiation group. 

Overall, the fish gelatin/sodium alginate/MAA composite film showed excellent anti-ultraviolet effects, which provide a theoretical basis and new ideas for research on the anti-ultraviolet properties of biological packaging films.

## 4. Conclusions

In this study, the extraction rates and antioxidant properties of MAAs from Rizhao porphyra, Quanzhou porphyra and Ningde porphyra were compared. It was observed that MAAs extracted from Ningde porphyra had the best antioxidant activity. The active composite films were successfully prepared by incorporating MAAs with fish gelatin and sodium alginate. The physicochemical properties, the antioxidant properties and antimicrobial activities of the composite films were determined. The results show that the composite films had a good appearance with a soft texture, uniform thickness, smooth and flat surface, uniform color and high transparency. The moisture content, solubility, swelling properties and water vapor permeability of the composite films did not change significantly with the change in the amount of MAAs added. Compared with the control films, MAAs improved antioxidant activities, UV resistance and the antibacterial properties of the films. Therefore, FG/SA composite films incorporated with MAAs possess promising potential for active food packaging in the food industry.

## Figures and Tables

**Figure 1 polymers-14-02980-f001:**
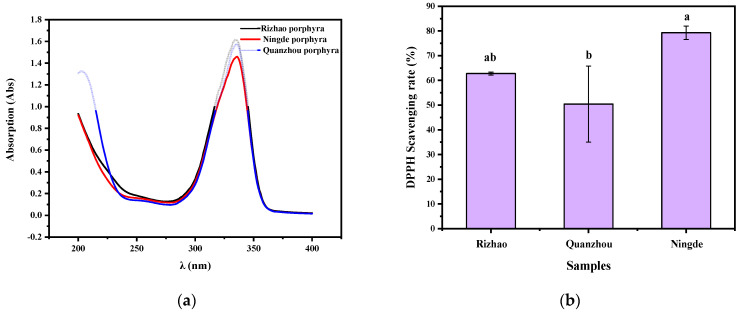
(**a**) The maximum absorption wavelength of extractions in the UV region. (**b**) DPPH free radical scavenging activities of MAAs.

**Figure 2 polymers-14-02980-f002:**
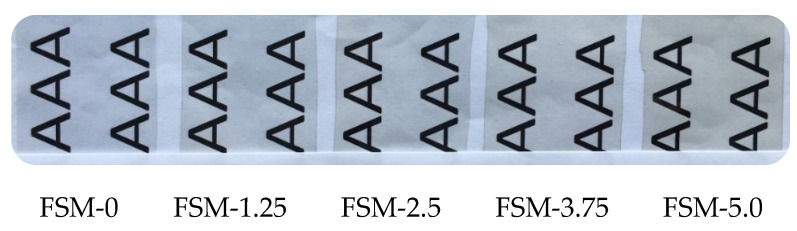
Composite edible film sample with different MAA concentrations.

**Figure 3 polymers-14-02980-f003:**
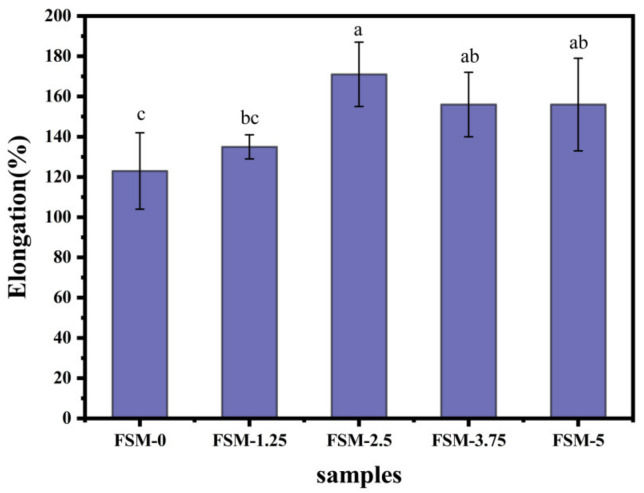
The impact of MAA concentrations on the elongation of composite films.

**Figure 4 polymers-14-02980-f004:**
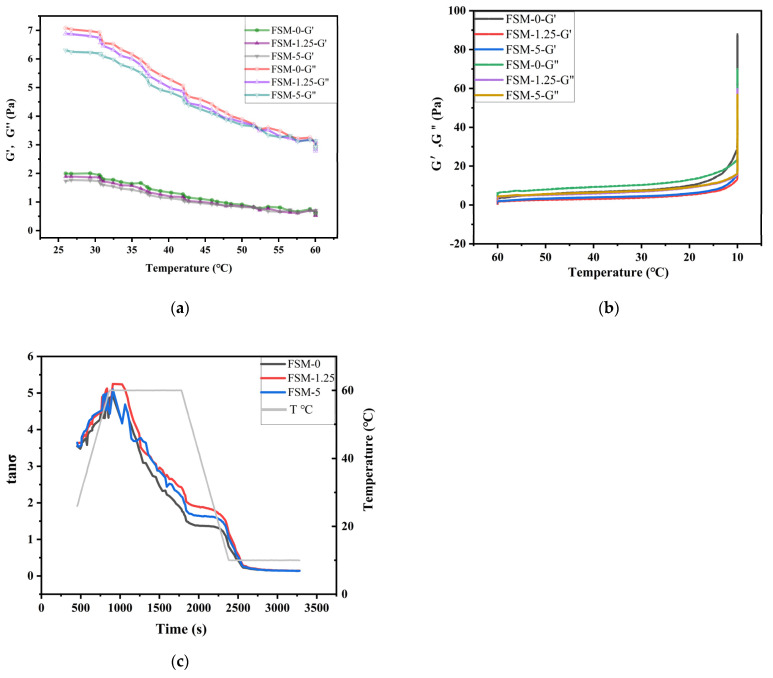
Temperature sweep curves of composite films: (**a**) Changes in storage modulus G′ and loss modulus G″ during heating regime; (**b**) Changes in G′ and G″ during cooling regime; (**c**) Changes in tan δ values in the temperature sweep process.

**Figure 5 polymers-14-02980-f005:**
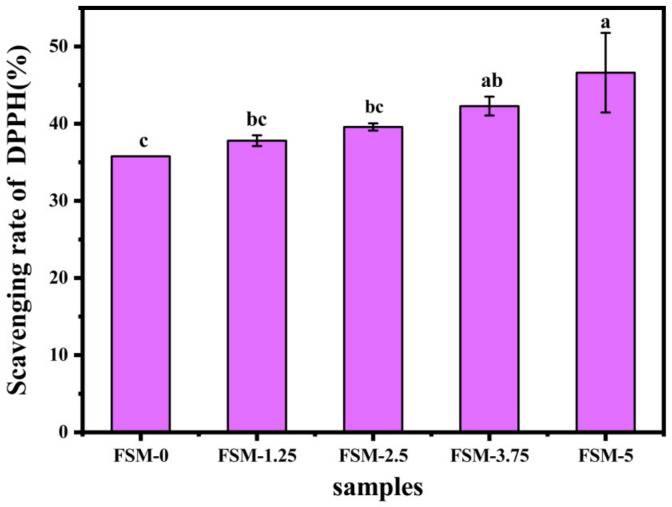
DPPH free radical scavenging activities of composite films.

**Figure 6 polymers-14-02980-f006:**
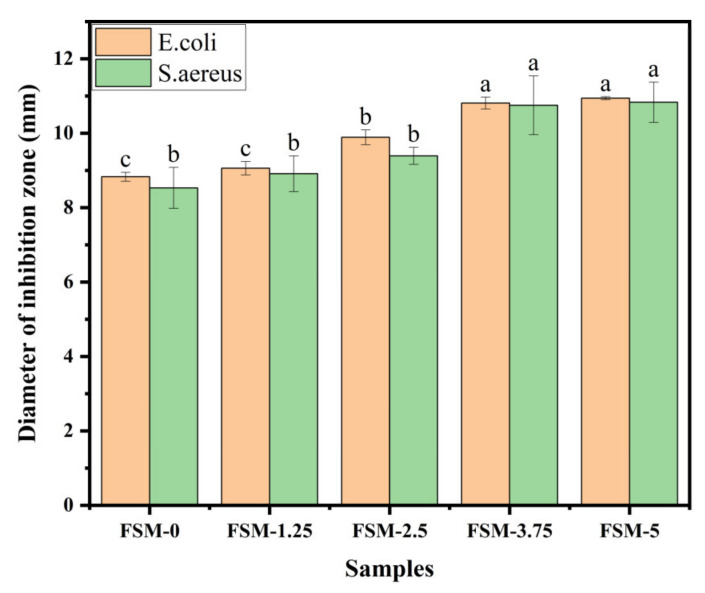
Antibacterial activity of composite films incorporated with different MAA ratios.

**Figure 7 polymers-14-02980-f007:**
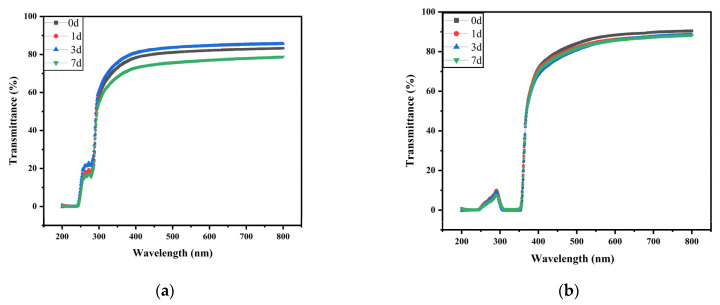
(**a**) Influence of storage time on light transmittance of FSM-0. (**b**) Influence of storage time on light transmittance of FSM-5.

**Figure 8 polymers-14-02980-f008:**
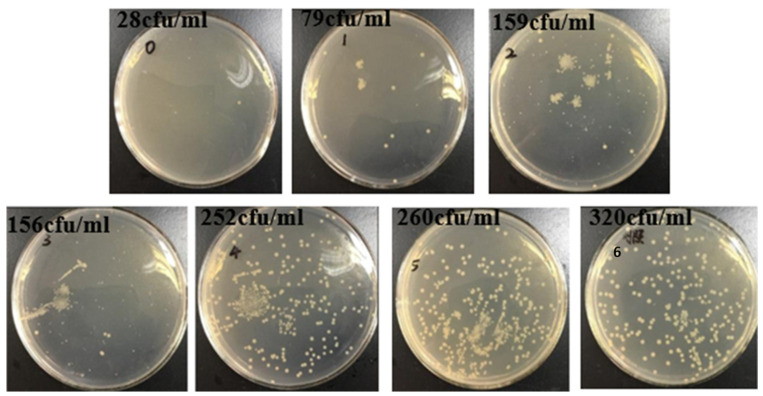
Survival of E. coli after 15 min of UV irradiation (0: uncoated, 1: FSM-0, 2: FSM-1.25, 3: FSM-2.5, 4: FSM-3.75, 5: FSM-5, 6: non-UV irradiation group).

**Table 1 polymers-14-02980-t001:** Effect of MAA concentrations on the chroma value of composite films.

Samples	L	a	b	ΔE
FSM-0	74.21 ± 1.79 ^a^	0.25 ± 0.17 ^c^	0.73 ± 0.19 ^d^	0.00
FSM-1.25	68.24 ± 0.77 ^b^	0.29 ± 0.12 ^c^	2.22 ± 0.37 ^c^	6.15
FSM-2.5	67.96 ± 1.97 ^b^	0.36 ± 0.18 ^bc^	2.30 ± 0.11 ^c^	6.45
FSM-3.75	67.32 ± 3.78 ^b^	0.62 ± 0.07 ^b^	3.70 ± 0.28 ^b^	7.51
FSM-5	64.76 ± 5.23 ^b^	1.30 ± 0.15 ^a^	6.03 ± 0.50 ^a^	10.89

Note: The same superscript letters indicate no significant difference (*p* > 0.05), and different superscript letters indicate significant differences (*p* < 0.05). Results are mean ± SD (n = 3).

**Table 2 polymers-14-02980-t002:** Water content, solubility, swelling ability and WVP of composite films.

Samples	Water Content(%)	Solubility(%)	Swelling Ability	WVP(g·mm/m^2^·h·kPa)
FSM-0	7.01 ± 0.43 ^a^	24.3 6± 5.25 ^b^	23.65 ± 3.40 ^a^	0.01742 ± 0.00047 ^a^
FSM-1.25	7.18 ± 0.40 ^a^	30.50 ± 1.83 ^ab^	25.13 ± 11.41 ^a^	0.01575 ± 0.00119 ^a^
FSM-2.5	6.05 ± 0.91 ^a^	27.73 ± 5.37 ^ab^	23.81 ± 4.09 ^a^	0.01780 ± 0.00249 ^a^
FSM-3.75	6.44 ± 0.93 ^a^	29.79 ± 0.51 ^ab^	29.26 ± 5.60 ^a^	0.01733 ± 0.00348 ^a^
FSM-5	6.35 ± 0.77 ^a^	32.90 ± 1.42 ^a^	23.12 ± 8.62 ^a^	0.01760 ± 0.00156 ^a^

Each value was the average of three repetitions and the standard deviation. There was no significant difference in the same column (*p* > 0.05)

**Table 3 polymers-14-02980-t003:** Light transmission of composite edible films.

Samples	Light Transmission (%) of Samples at Different Wavelengths (nm)					
	200	280	350	400	600	800
FSM-0	0.70	15.10	74.30	80.80	85.30	86.50
FSM-1.25	0.50	6.70	2.30	75.70	84.30	86.10
FSM-2.5	0.60	5.85	1.45	72.60	85.15	87.25
FSM-3.75	0.40	4.28	1.38	71.15	84.83	86.68
FSM-5	0.30	5.00	0.60	69.50	85.80	88.40

## Data Availability

The data presented in this study are available on request from the corresponding author.

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
