# Peer review of "The Preparation of Anti-Ultraviolet Composite Films Based on Fish Gelatin and Sodium Alginate Incorporated with Mycosporine-like Amino Acids"

_polymers, 2022, doi:10.3390/polym14152980_

Round 1

Reviewer 1 Report

The manuscript is a good and novel research performed by the team. however, there are a few comments below to improve the quality of it:

- The abstract is not representative it should be rewritten. the abstract should be strong enough to attract the readers and be representative 

-Line 51: It would be better to mention the shortcoming of sodium alginate as well not just fish gelatin, so that you can say it is good to combine them.

-Line 93: "to the best of our knowledge, FG/SA composite films containing MAAs have never been investigated." delete this because it has been repeated in line 96

- Line 102: was also determined.

-Line 109: local company.?? the name of the company, city ??

-Line 119: was calculated

-Section 2.3.1. and 2.3.2: : Authors have analyzed the UV scanning of MAAs with the concentration of 0.1 g/L, however, in the section 2.3.2. it is 10 times more for DPPH radical scavenging activity. On the other hand the concentration of the MAAs in the film composite is way different (0-5% of FG)?? How did the authors determined these concentrations at all?

-Line 134: 4% v/v?? Was the gelatin powder or not? if not, then what was the initial concentration??

- There is no need to bring Table 1 because the only changing concentration is the MAAs which is mentioned in the text. It will be better to just mention the concentration of sodium alginate and glycerol (1 and 3 %) in the text as well.

- Section 3.1.: Needs more discussion and bringing some relevant studies 

-Line 242: It would be of the best if authors can state that what is the most acceptable appearance for the consumers. this will justify the reason of doing this experiment.

-Line 243: the color of edible film is often used

-Line 273: "The high water content (more than 30%) made unfavorable to the 273 storage of the film and most food products" ??can you explain this? and please check the English

-Line 277: "  It can be seen that the film can be degraded" !? you mean because of the growing rate of the solubility it would be degraded? How is this an advantage? what if this film become solubilized in food water which is in the packaging?

- Conclusions section: "properties of MAAs were compared" The authors did not compare the MAAs properties with any other compound. Please just rewrite this .

- According to the results of the antibacterial activity, the MAAs had significant activity until 3.75 %. So why didn't the author mention that properly in the conclusion? the insist is on the antioxidant and UV resistance

Reviewer 2 Report

The manuscript is innovative enough, with the correct state-of-the-art and innovative approach related to different aspects describing the preparation of anti-UV packaging films with MAAs. The Materials and Methods, together with the Results are convincingly presented, with possible positive impact that might provide a novel potential candidate material for food preservation and broaden the application of MAAs in the food industry.

However, there are some important mistakes within the text (few examples):

- row 27-28): Overall, this project provides a theoretical basis for the study of active composite films with anti-ultraviolet(...? activities?properties?), and provides new ideas for the application of MAAs. (similar thing in the row 66)

-row 88: the sentence is not clear, and should be rewritten or more precisely taken from the original source (MAAs are enriched in red algae,...)

To be rewritten:

-the row 214 (After each medium was exposed to the...) and 216 (Take the uncoated flat plate and the ....)

- several misspellings among entire text (row 43, row 198, 328, properties; 277: font size;
